# Novel Insight of N6-Methyladenosine in Cardiovascular System

**DOI:** 10.3390/medicina61020222

**Published:** 2025-01-26

**Authors:** Huan Zhang, Wei Lu, Haoyue Tang, Aiqun Chen, Xiaofei Gao, Congfei Zhu, Junjie Zhang

**Affiliations:** 1Department of Cardiology, Nanjing First Hospital, Nanjing Medical University, Nanjing 210006, China; zhang07182022@163.com (H.Z.); xmqwo07@163.com (W.L.); jsdfthy@163.com (H.T.); yeqicaq@163.com (A.C.); gaoxiaofei2014@163.com (X.G.); 2Department of Cardiology, Lianshui County People’s Hospital, Affiliated Hospital of Kangda College, Nanjing Medical University, Huaian 223400, China

**Keywords:** N6-methyladenosine, cardiovascular system, ischemia–hypoxia injury, atherosclerosis, heart failure

## Abstract

N6-methyladenosine (m6A) is the most common and abundant internal co-transcriptional modification in eukaryotic RNAs. This modification is catalyzed by m6A methyltransferases, known as “writers”, including *METTL3/14* and *WTAP*, and removed by demethylases, or “erasers”, such as *FTO* and *ALKBH5*. It is recognized by m6A-binding proteins, or “readers”, such as *YTHDF1/2/3*, *YTHDC1/2*, *IGF2BP1/2/3*, and *HNRNPA2B1*. Cardiovascular diseases (CVDs) are the leading cause of morbidity and mortality worldwide. Recent studies indicate that m6A RNA modification plays a critical role in both the physiological and pathological processes involved in the initiation and progression of CVDs. In this review, we will explore how m6A RNA methylation impacts both the normal and disease states of the cardiovascular system. Our focus will be on recent advancements in understanding the biological functions, molecular mechanisms, and regulatory factors of m6A RNA methylation, along with its downstream target genes in various CVDs, such as atherosclerosis, ischemic diseases, metabolic disorders, and heart failure. We propose that the m6A RNA methylation pathway holds promise as a potential therapeutic target in cardiovascular disease.

## 1. Introduction

More than 140 types of chemical modifications have been discovered in RNA. Among these modifications, RNA methylation involves the addition of methyl groups to specific nucleotide residues in RNA, such as N1-methyladenosine (m1A), N6-methyladenosine (m6A), N6,2′-O-dimethyladenosine (m6Am), and 2′-O′-methylation (2′-OMe) [1,2,3]. Among these modifications, the methylation of mRNA at the N-6th position of the adenosine residue (m6A) has been extensively studied since its discovery in the 1970s. It is the most common methylation modification found in the internal sequence of mRNA in eukaryotes, and it is the most abundant and evolutionarily conserved reversible post-transcriptional modification [4,5,6]. Functionally, m6A is involved in various aspects of RNA biology, including RNA stability, nuclear export, translation efficiency, and mRNA degradation [2,3,7]. In addition to regulating RNA coding sequences (mRNA), m6A methylation also plays a regulatory role in non-coding RNAs such as microRNAs, long non-coding RNAs, and circular RNAs, thereby influencing gene expression in diverse physiological processes [8,9,10].

Cardiovascular diseases (CVDs) are the leading cause of morbidity and mortality worldwide, and their pathological mechanisms are complex [11]. Despite significant advances in the diagnosis, treatment, and prognosis of CVDs, there is an urgent need for new diagnostic biomarkers and treatments to reduce mortality and improve therapeutic outcomes for patients with CVDs.

## 2. m6A RNA Methylation

The rapid advancement of bioinformatic analyses and high-throughput sequencing technologies has revealed that m6A modification primarily occurs at the consensus motif RR (m6A) CH (R = A or G, H = A, C, or U) in long internal exons, near stop codons, or in the 3′ untranslated region (UTR) [7,12] (Figure 1). Importantly, m6A modification is dynamic and its methylation levels undergo dynamic changes, regulated by various methyltransferases (“writers”), m6A-binding proteins (“readers”), and demethylases (“erasers”) [13,14]. These three groups of proteins are responsible for the addition, removal, and recognition of m6A RNA modification.

Studies have shown that m6A modification can alter RNA secondary structures or be recognized by “readers” to regulate the metabolism of methylated mRNAs, thereby exerting various functions [15]. The process of RNA modification by m6A begins during transcription. The addition and removal of m6A mostly occur in the nucleus, where specific nuclear “readers” can bind to m6A and potentially affect mRNA splicing and other nuclear processes. After export to the cytoplasm, m6A interacts with specific cytoplasmic reader proteins that influence RNA stability, translation efficiency, and localization [2] (Table 1).

M6A “writers” consist of *METTL3/14*, *KIAA1429*, WTAP, zinc finger CCCH domain-containing protein 13 (*ZC3H13*), and *RBM15/15B* [31,32]. Among these, *METTL3* and *METTL14* are considered the most critical core components of the m6A methyltransferase complex [16]. *METTL3* acts as the catalytic subunit, while *METTL14* plays a structural role and activates *METTL3* through allosteric and RNA substrate recognition [16,33]. WTAP interacts with *METTL3*-*METTL14* and is the third subunit of the complex [17,18]. *RBM15/15B* is responsible for recruiting the methylation complex to *XIST* [19]. *ZC3H13* plays a critical role in anchoring *WTAP*, Virilizer, and Hakai in the nucleus for m6A methylation regulation and mESC self-renewal [20]. *KIAA1429*, as a top interactor with *WTAP*, recruits specific cleavage and polyadenylation specificity factors, leading to a longer 3′UTR selection [21]. ZC3H13, *KIAA1429*, and *RBM15/RBM15B* play key roles in localizing the methyltransferase complex in nuclear speckles and U-rich regions adjacent to m6A sites in mRNAs.

*FTO* (FAT mass and obesity-associated protein) and *ALKBH5* (ALKB homolog 5 protein) belong to the ferric divalent/α-ketoglutarate dependent dioxygenase ALKB protein family and exhibit efficient oxidative demethylation activity targeting N6-methyladenosine residues in RNA [2,23]. Both *FTO* and *ALKBH5* are located in the nuclear speckle area, bind to m6A-modified RNA, catalyze the oxidation of m6A to adenine, and produce the intermediate products hm6A (N6 hydroxymethyl adenosine) and further oxidation product f6A (N6 formyl adenosine).

“Readers” are proteins that recognize m6A methylation sites on RNAs, bind to mRNA, and influence nuclear export, RNA stability, or degradation. The YTH domain-containing proteins, including *YTHDC1*, *YTHDC2* (DC2), *YTHDF1* (DF1), DF2, and DF3, were the first m6A-binding proteins to be discovered [14]. *YTHDF1* promotes mRNA translation, while *YTHDF2* accelerates the decay of m6A-modified transcripts. The role of *YTHDF3* is complex, as it facilitates *YTHDF1*’s role in promoting translation, and knockdown of *YTHDF3* results in reduced translation efficiency of mRNA targets of both *YTHDF3* and *YTHDF1* [24,25]. A recent study showed that *YTHDC1* blocks *SRSF10* mRNA binding by selectively recruiting the *SRSF3* mRNA pre-splicing factor, thereby promoting exon inclusion in the target transcript [34]. Nuclear export of m6A-methylated transcripts may be facilitated by *YTHDC1* through interaction with nuclear transport receptors [27]. *YTHDC2* is an N6-methyladenosine-binding protein that regulates mammalian spermatogenesis by enhancing the translation efficiency of its targets and also decreasing their mRNA abundance [28]. IGFBP proteins, including *IGF2BP1/2/3*, can regulate RNA localization, translation, and stability [29]. The heterogeneous nuclear ribonucleoprotein (hnRNP) family, including *hnRNPG*, *hnRNPC*, and *hnRNPA2B1*, is another group of m6A “reader” proteins. Wu et al. found that m6A can enhance the ability of *hnRNPA2/B1* to enhance nuclear events, such as pri-miRNA processing [30].

## 3. Role of m6A in Cardiovascular Disease

### 3.1. Risk Factors Associated with CVDs

Unhealthy diet, obesity, and diabetes are well-recognized risk factors involved in the pathophysiology of numerous chronic cardiovascular diseases (CVDs). In this comprehensive review, we aim to summarize the current state of research regarding the mechanisms underlying m6A methylation and its role in the development of these CVD risk factors. By examining the existing literature, we can gain insights into the molecular processes driving m6A methylation and its contribution to the pathogenesis of CVDs. Importantly, these findings offer valuable information that can guide future therapeutic interventions and help identify potential targets for the prevention and management of CVDs. Table 2 provides a summary of the key findings and future therapeutic prospects in this field.

#### 3.1.1. Glucose Metabolism

The regulation of glucose metabolism is crucial in maintaining cardiovascular health, as diabetes or hyperglycemia can have detrimental effects on the cardiovascular system. Correcting or preventing abnormalities in glucose metabolism can significantly reduce the incidence of cardiovascular disease. Yang et al. demonstrated that high glucose stimulation enhanced the expression of *FTO*, which resulted in decreased m6A methylation. *FTO* then triggered the mRNA expression of *FOXO1* and *G6PC*, which are associated with gluconeogenesis [35]. This finding suggests that controlling blood glucose levels by regulating *FTO* expression could be a potential method for managing glucose metabolism.

Additionally, *FTO* plays a regulatory role in improving glucose levels and may have implications in protecting the injured heart or improving heart function in heart failure patients. In a recent study, a team created a model of transverse aortic constriction-induced heart failure (HF) in mice and found that cardiac fibrosis and hypertrophy were ameliorated in mice overexpressing *FTO*. They further observed that 18F-FDG uptake, an indicator of glucose uptake, was significantly increased in mice overexpressing *FTO*, confirming *FTO*’s role in regulating glucose uptake and glycolysis for the attenuation of cardiac dysfunction [41].

The depletion of m6A levels in pancreatic β-cells was found to induce cell-cycle arrest and impair insulin secretion by decreasing *AKT* phosphorylation and *PDX1* protein levels, as reported by De Jesus et al. [36]. Another study by Krüger et al. in 2019 supported this concept, demonstrating that the loss of endothelial *FTO* protected mice from high-fat-diet-induced glucose intolerance and insulin resistance by increasing *AKT* phosphorylation in endothelial cells and skeletal muscle [37].

#### 3.1.2. Adipogenesis and Obesity

The deposition of lipids in the arterial intima and the formation of foam cells are closely associated with adipogenesis. Obesity, being a major risk factor for coronary heart disease, necessitates effective control of adipogenesis to prevent atherosclerosis and subsequent coronary syndromes such as myocardial infarction.

Current evidence strongly supports the role of *FTO* in promoting adipogenesis. *KLF4*, an essential early regulator of adipogenesis, is normally suppressed by *C/EBPβ* through a negative feedback loop involving *Krox20* and *KLF4* expression [42]. Wu et al. found that *FTO* depletion in porcine and mouse preadipocytes inhibited adipogenesis through the *JAK2*-*STAT3*-*C/EBPβ* signaling pathway [38]. Zhao et al. revealed that *FTO* controls the exonic splicing of *RUNX1T1*, an adipogenesis-related transcription factor, by regulating m6A levels around splice sites, thereby promoting adipogenesis [39]. The relationship between *FTO* and autophagy remains controversial. Wang et al. demonstrated that *FTO* knockdown inhibited autophagy and adipogenesis by decreasing the expression of ATG5 and *ATG7*, providing a new mechanism to inhibit adipogenesis [40]. Furthermore, in a study conducted by Song et al., it was verified that Zinc finger protein 217 (*Zfp217*), a known oncogenic protein, promotes adipogenesis through m6A mRNA methylation via *FTO* and *YTHDF2*. Interestingly, the study also highlighted that *YTHDF2* blocks the demethylase activity of *FTO*, leading to increased m6A levels, while *Zfp217* functions as a regulator to maintain *FTO*’s m6A demethylation activity [43]. These findings suggest that *FTO* knockdown or *YTHDF2* overexpression could serve as potential targets for obesity therapy aimed at the prevention of cardiovascular diseases.

### 3.2. Function of m6A in CVDs

Currently, numerous studies have investigated the role of m6A in cardiovascular diseases (CVDs), including ischemia–hypoxia injury, atherosclerosis, acute myocardial infarction (AMI), and heart failure (Table 3, Figure 2). The effects of m6A modification vary across different CVDs. In this review, we aim to explore the relationship between m6A and CVDs in order to identify potential promising therapeutic targets.

#### 3.2.1. m6A and Ischemia–Hypoxia Injury

Ischemia and hypoxia are recognized as common causes of vascular injury. *TFEB* is a master regulator of lysosomal biogenesis and autophagy genes. In cardiomyocytes subjected to hypoxia–reoxygenation, the increased expression of *METTL3* regulates the m6A modification of *TFEB* mRNA by promoting *HNRNPD* association with *TFEB* pre-mRNA [46]. This dysregulation impairs autophagic flux and enhances apoptosis in hypoxia–reoxygenation-treated cardiomyocytes. Conversely, *ALKBH5* reverses the hypoxia–reoxygenation-mediated m6A modification of *TFEB* mRNA in cardiomyocytes, providing a potential mechanism for cellular prognosis regulation by *METTL3* through autophagy modulation under ischemic and hypoxic conditions. Apart from *TFEB*, *METTL3* has also been shown to promote *DGCR8* binding to pri-miR-143-3p, thereby enhancing miR-143-3p expression and inhibiting the transcription of *PRKCE*, a gene implicated in autophagy, which further exacerbates cardiomyocyte pyroptosis and myocardial ischemia–reperfusion injury [69].

Interestingly, there is evidence suggesting contrasting roles of *METTL3* and *ALKBH5* in response to ischemia–hypoxia injury. While *METTL3* exacerbates injury to cells under ischemic–hypoxic conditions, *ALKBH5* contributes to angiogenesis maintenance in endothelial cells following acute ischemic stress by reducing *SPHK1* m6A methylation and downstream *eNOS*-*AKT* signaling [47]. In addition, *ALKBH5* overexpression in cardiomyocytes leads to increased translation of *YAP*, which undergoes *YTHDF1*-mediated m6A modification, enhancing the proliferative capacity of human and mouse cardiomyocytes [48]. However, it appears that *ALKBH5* may sometimes hinder angiogenesis. Zhao et al. demonstrated that *ALKBH5* overexpression attenuates blood flow recovery and angiogenesis post-ischemic injury by promoting *WNT5A* mRNA decay and decreasing its half-life [49]. These contradictory findings suggest that *ALKBH5* may have different roles in distinct stages of disease progression or during different periods of human development. Further investigation is needed to elucidate the role of m6A in cardiovascular diseases, particularly in vascular injury.

*ULK1*, serving as a protein kinase activated upon autophagy stimulation, plays a critical role in recruiting other autophagy-related proteins to the autophagosome formation site [70]. Jin et al. demonstrated in a 2018 study that *FTO* demethylates *ULK1* transcripts, thereby prolonging the half-life of *ULK1* transcripts and promoting autophagy while inhibiting cellular apoptosis [71]. Similarly, *FTO* overexpression inhibits apoptosis in hypoxia–reoxygenation-treated myocardial cells through the regulation of m6A modification in *Mhrt* [50]. Additionally, Tao Yin et al. found that exosome-based *WTAP* siRNA delivery ameliorated myocardial ischemia–reperfusion injury by influencing the m6A modification of *TXNIP* mRNA [51].

#### 3.2.2. m6A and Atherosclerosis

##### m6A in Vascular Smooth Muscle Cell (VSMC) Differentiation and Angiogenesis

VSMC differentiation is a common phenotype observed in various cardiac vascular diseases, particularly in atherosclerosis [72]. One study investigated the promotion of the differentiation of adipose-derived stem cells (ADSC) into VSMCs under hypoxic stress. The expression pattern of *METTL3* in their experiment was consistent with that of VSMC-specific markers, such as *α-SMA*, *SM22α*, and calponin, suggesting an important role of *METTL3* in VSMC differentiation and potential changes in VSMCs in CVDs [73]. Additionally, circ*YTHDC2*, a non-coding endogenous RNA, has been shown to promote the dedifferentiation of VSMCs into a “synthetic type” through the regulation of *TET2* expression. The m6A modification mediated by *YTHDC2* stabilizes circ*YTHDC2*, highlighting the potential of the *YTHDC2/circYTHDC2/TET2* pathway as an important target to inhibit atherosclerosis through dedifferentiation methods [74].

Apart from its role in differentiation, m6A can also mediate the process of angiogenesis. Parial et al. observed that *METTL3* activates the phosphorylation of *PHLPP2*-*mTOR*-*AKT* signaling to enhance angiogenesis [75]. Furthermore, *METTL3* enhances the translation of *LRP6* and *DVL1* in a *YTHDF1*-dependent manner, thereby regulating the *Wnt* signaling pathway to exert its angiogenic role [76]. These studies suggest that targeting m6A modification holds promise as a strategy for the treatment of angiogenic diseases.

##### m6A and Calcification

Calcification is frequently observed in atherosclerosis and valvular diseases, often leading to plaque rupture, thrombosis, valve sclerosis, and valve insufficiency, which can result in serious complications like heart failure. Zhou et al. reported that *METTL3* is highly expressed in human calcified aortic valves compared to normal valves. Mechanistically, *METTL3* promotes the osteogenic differentiation of human aortic valve interstitial cells by suppressing *TWIST1* expression in an m6A-*YTHDF2*-dependent manner [44]. Additionally, Chen et al. demonstrated that decreasing *METTL14* expression in calcified arteries attenuates indoxyl sulfate-induced m6A modification and reduces human artery smooth muscle cell (HASMC) calcification. Knockdown of *METTL14* may enhance vascular repair function by reducing calcification, presenting a potential therapeutic approach for atherosclerosis and valvular diseases [45].

##### m6A in Atherosclerosis

Atherosclerosis, the leading cause of CVD, has a high mortality rate among the affected population. The involvement of inflammation in the pathogenesis of atherosclerosis and its complications has gained considerable attention [77]. Chien et al. discovered that oscillatory stress (OS) up-regulates *METTL3* expression, leading to increased *NF-κB* p65 Ser536 phosphorylation and enhanced monocyte adhesion, thus promoting atherosclerosis. Mechanistically, *METTL3*-mediated hypermethylation stabilizes *NLRP1* mRNA while inducing the degradation of *KLF4* mRNA under OS. The m6A hypermethylation is recognized by *YTHDF1* and *YTHDF2* reader proteins, respectively [52]. Another study investigating the effect of m6A methylation on endothelial cells showed that *METTL3* is highly expressed in oxidized low-density lipoprotein (ox-LDL)-induced dysregulated human umbilical vein endothelial cells (HUVECs). *METTL3* knockdown prevents the progression of atherosclerosis by inhibiting the *JAK2/STAT3* pathway via *IGF2BP1* [54]. These findings provide a basis for targeting *METTL3* molecules and their downstream pathways to delay the progression of atherosclerosis.

Inflammation increases blood monocyte adhesion and migration into the subendothelial space, which is a critical event in the development of atherosclerosis [78]. *METTL14*, induced by *TNF-α*, promotes *FOXO1* expression, leading to the increased transcription of *VCAM-1* and *ICAM-1* by enhancing m6A modification and inducing the endothelial cell inflammatory response and atherosclerotic plaque formation [53]. Moreover, *METTL14* promotes *DGCR8*-mediated processing of pri-miR-19a, leading to the formation of mature miR-19a in atherosclerotic vascular endothelial cells (ASVECs), thereby promoting the proliferation and invasion of ASVECs [55]. Therefore, *METTL14* plays a major role in endothelial cell activation and represents a novel therapeutic target.

The role of VSMCs in atherosclerosis has been extensively studied, highlighting their capability to switch to transitional, multipotential cells adopting various cellular states, including inflammation, ossification, and collagen matrix deposition [79]. Recently, it has been shown that m6A modification is involved in the regulation of VSMC behavior. For instance, dihydroartemisinin alleviates angiotensin II-induced VSMC proliferation and inflammatory response by blocking the *FTO/NR4A3* axis [56]. TPNS inhibits VSMC proliferation, migration, and intimal hyperplasia by regulating the *WTAP/p16* signaling pathway [57].

#### 3.2.3. m6A and Acute Myocardial Infarction

Coronary atherosclerosis is a chronic disease characterized by stable and unstable periods. During unstable periods, when inflammation is activated in the vascular wall, patients may experience myocardial infarction (MI). Inflammation plays a fundamental role in atherogenesis and the pathophysiology of ischemic events [80].

M1 macrophages exhibit robust antimicrobial and antitumoral activity, but they also mediate tissue damage induced by reactive oxygen species (ROS), impairing tissue regeneration and wound healing [81]. Liu et al. suggested that *METTL3* directly methylates *STAT1*, the master transcription factor controlling M1 macrophage polarization [82]. The inflammatory pathways mediated by *METTL3* are not limited to a single mechanism. In Wang’s study, Mettl3-mediated m6A modification of *CD40*, *CD80*, and *TLR4* signaling adaptor Tirap transcripts enhanced their translation in dendritic cells, strengthening *TLR4/NF-κB* signaling-induced cytokine production [83]. Therefore, *METTL3* can serve as an anti-inflammatory target during myocardial infarction to reduce tissue damage. Regarding tissue recovery after an ischemic event, *METTL3* knockdown may be a feasible approach. Loss of *METTL3*/m6A impairs the maturation of pri-miR-143, resulting in low expression of miR-143-3p, which in turn leads to high expression of *Yap* and *Ctnnd1*. This promotes cardiomyocyte proliferation and endogenous heart regeneration after MI [58].

We must not overlook the tissue damage caused by ischemia–reperfusion (I/R) injury, which is a significant cause of death in MI. Such damage occurs when blood supply is restored, and excessive free radicals attack the tissues. Chen’s study showed a positive correlation between the expression of *ALKBH5* in the infarct area and the levels of *CK-MB* and *LDH* in peripheral blood. Additionally, *ALKBH5* was found to regulate the content of amino acids involved in the tricarboxylic acid (TCA) cycle and mediate the activity of key enzymes, thereby affecting cell metabolism and survival [59]. Moreover, *ALKBH5* has been shown to activate the *EGFR-PI3K-AKT-mTOR* signaling pathway, enhancing the stability of *BCL-2* mRNA and promoting the interaction between *Bcl-2* and *Beclin1*, thereby inhibiting autophagy [84]. Kun Yang et al. demonstrated that *ALKBH5* promotes the stability of *ErbB4* mRNA and the degradation of *ST14* mRNA through m6A demethylation, leading to improved fibroblast-to-myofibroblast transformation and better post-MI repair [60]. Considering the role of *ALKBH5* in MI, inhibiting *ALKBH5* could be a potential method to improve heart dysfunction after MI.

m6A is also involved in tissue repair after ischemic events. The reduced expression of *METTL14* in human artery smooth muscle cells (HASMCs) decreases calcification and enhances vascular repair function [45]. However, Pang et al. found that *METTL14* attenuates cardiac ischemia–reperfusion injury by increasing the m6A modification of *Wnt1* mRNA, leading to up-regulation of the *Wnt1* protein and subsequent activation of the *Wnt1/β-catenin* signaling pathway [61]. In fact, N6-methyladenosine and the *Wnt* protein family have close interactions. In a recent study in 2021, *ALKBH5* was found to decrease the stability of *WNT5A*, thereby impeding angiogenesis in hypoxic cardiac microvascular endothelial cells [49].

#### 3.2.4. m6A and Heart Failure (HF)

Heart failure occurs as a consequence of abnormalities in cardiac structure, function, rhythm, or conduction [85]. Despite improvements in survival rates, the absolute mortality rates for HF remain around 50% within 5 years of diagnosis [86]. Researchers have begun exploring the potential of m6A-mediated approaches for improving heart failure. Zhang et al. found that the expression of *METTL3*, *METTL4*, *KIAA1429*, *FTO*, and *YTHDF2* was significantly up-regulated in patients with heart failure with preserved ejection fraction (HFpEF) compared to healthy controls, suggesting that m6A methylation may be a therapeutic target for HFpEF interventions [87].

In addition to restoring contractile protein expression, such as *SERCA2*, which can alleviate cardiac insufficiency, Mathiyalagan et al. provided experimental evidence that *FTO* overexpression decreased cardiac fibrosis and enhanced angiogenesis in the ischemic myocardium [62]. This conclusion aligns with the results of Berulova et al., who analyzed m6A RNA methylation through next-generation sequencing and found alterations in the m6A landscape in heart hypertrophy and heart failure [88]. Furthermore, they demonstrated that cardiomyocyte-specific *FTO* knockdown mice exhibited impaired cardiac function compared to control mice. Hence, the role of *FTO* is closely associated with recovering cardiac function. In a recent 2024 study, macrophage-specific knockout of *ALKBH5* inhibited angiotensin II-induced macrophage-to-myofibroblast transition, subsequently ameliorating cardiac fibrosis and dysfunction. This mechanism involves the targeting of interleukin-11 (*IL-11*), leading to increased *IL-11* mRNA stability and protein levels [65].

Contrary to demethylation, although *METTL3* is positively related to angiogenesis [75,76], it serves as a negative regulator in HF. Hinger et al. found increased levels of m6A and *METTL3* in human non-ischemic failing hearts [89]. In a recent study, silencing *METTL3* reduced m6A modification levels in fibrosis-related genes and decreased myocardial fibrosis in mice with myocardial infarction. This was achieved by inhibiting proliferation, fibroblast-to-myofibroblast transition, and collagen accumulation [90]. Maslinic acid has been demonstrated to protect against pressure-overload-induced cardiac hypertrophy by blocking *METTL3*-mediated m6A methylation [91]. A study by Dorn et al. identified enhanced *METTL3*-mediated methylation of mRNA on N6-adenosines in response to hypertrophic stimuli, driving cardiomyocyte hypertrophy in vitro and in vivo [92]. However, Kmietczyk’s experiments with *METTL3*-overexpressing mice and control mice subjected to transverse aortic constriction surgery to induce pathological hypertrophy led to a contrasting finding—pathological hypertrophic cellular growth was attenuated in the hearts of *METTL3*-overexpressing mice [93]. Kmietczyk explained this contradiction by suggesting that Dorn et al. used a transgenic model in the FVB background, while the in vivo studies utilized a C57Bl6/N background, and various methods were employed to overexpress *METTL3*.

In Xu et al.’s study, *YTHDF2* was also found to increase in human HF samples. Moreover, *YTHDF2* protein levels increased in cell and animal models of cardiac hypertrophy stimulated by isoproterenol or phenylephrine. Mechanistically, *YTHDF2* recognizes the m6A site on *Myh7* (beta-myosin heavy chain) mRNA, promoting its degradation and thereby alleviating cardiac hypertrophy [63]. MiR-133a represses cardiac hypertrophy and hypoxia-induced apoptosis [94,95]. Qian et al. found that *IGF2BP2* binding to the m6A-modified site could enhance the accumulation of the miR-133a-*AGO2*-*RISC* complex on its targets, augmenting the repression effect of miR-133a and ultimately inhibiting cardiac hypertrophy [64].

#### 3.2.5. m6A and Other CVDs

The *FTO* gene is strongly expressed in the hypothalamus, a brain structure involved in blood pressure regulation. Marcadenti et al. demonstrated that common genetic variants of *FTO* rs9939609 are negatively associated with diastolic and mean blood pressure in men with hypertension [96]. Through large-scale genome-wide association studies, Mo et al. recently revealed the crucial role of m6A in blood pressure regulation. They identified 1236 m6A-SNPs, such as rs9847953 and rs197922, that are potentially associated with blood pressure levels [97]. However, the underlying mechanism of this association still needs further exploration.

In a study on hypoxia-mediated pulmonary hypertension (HPH), researchers identified a transcriptome-wide map of m6A circRNAs and found a reduction in m6A levels in circRNAs in lungs exposed to hypoxia. Furthermore, m6A influenced the circRNA-miRNA-mRNA co-expression network in hypoxia, involving *circXpo6* and *circTmtc3*, indicating the important role of m6A in HPH [98]. In a recent study using a rat model exposed to hypoxic conditions, increased *YTHDF2* recognition of *METTL3*-mediated m6A modification in *PTEN* mRNA promoted *PTEN* degradation via *PI3K/Akt* signaling, leading to the excessive proliferation of pulmonary artery smooth muscle cells (PASMCs) [66]. Additionally, *YTHDF1* was found to promote pulmonary hypertension by increasing the translational efficiency of *MAGED1*, which is involved in the hypoxia-induced proliferation of PASMCs through up-regulating *PCNA* [67]. These pieces of evidence suggest that down-regulating the level of N6-methyladenosine could be useful in the treatment of HPH.

In the context of aneurysms, similar to how *METTL14* promotes *DGCR8*-mediated processing of pri-miR-19a in atherosclerosis [69], *METTL3*-dependent m6A methylation promotes primary miR-34a maturation through *DGCR8*, leading to decreased *SIRT1* expression and an aggravated aneurysm formation [68]. This pathway represents a novel therapeutic target and diagnostic biomarker for aneurysm treatment. Vascular smooth muscle *FTO* promotes aortic dissecting aneurysms via m6A modification of *Klf5*, enhancing the *GSK3β* signaling pathway [99]. Moreover, Tan Li et al.’s genome-wide approaches revealed that aneurysm-associated m6A-SNPs might be linked to aneurysm pathogenesis by influencing local gene expression through m6A modification. They specifically identified two m6A-SNPs, NECTIN2 rs6859 and HPCAL1 rs10198139 [100].

#### 3.2.6. m6A in CVDs in the Era of Artificial Intelligence and Machine Learning

Contemporary methodologies that incorporate AI technology and chemosynthesis in drug discovery offer benefits including increased speed, user friendliness, and cost efficiency [101]. They applied a high-throughput fluorescence polarization assay to screen selective inhibitors of *FTO* from an older drug library containing 900 drugs and finally identified meclofenamic acid as an inhibitor of *FTO*.

As for in the field of CVDs, a multi-ethnic study of atherosclerosis represented that machine learning in conjunction with deep phenotyping improves prediction accuracy in cardiovascular event prediction in an initially asymptomatic population, and these methods may lead to greater insights into subclinical disease markers without a priori assumptions of causality [102]. In this work, Bharath et al. confirmed the influence of certain markers and risk factors on CV events, such as *TNF-α* SR and *IL2* SR. Such an encouraging study made people believe that machine learning may be useful to characterize cardiovascular risk, predict outcomes, and identify biomarkers.

At present, some teams have begun to combine m6A with artificial intelligence and machine learning to predict and diagnose the occurrence and development of CVDs. In 2021, Xu-shen Xiong et al. tried to excavate genetic drivers of m6A methylation in human tissues, including the heart and muscle [103]. First, they reported 129 transcriptome-wide m6A profiles, covering 91 individuals and 4 tissues (brain, lung, muscle, and heart) from GTEx/eGTEx. And they integrated m^6^A-QTLs with disease genetics, including heart/muscle m^6^A-QTLs underlying coronary artery disease. Then, they had some remarkable results: coronary-artery-disease-associated rs888298 is a muscle/heart m^6^A-QTL targeting the cardiac myocyte mitochondrial oxidation regulator *WIPI1* involved in signaling and autophagy. Heart-pulse-rate-associated rs6791834 is a muscle/heart m^6^A-QTL for myocyte microtubule differentiation regulator *MAP4* involved in heart development. High-blood-pressure-associated rs56104944 is a muscle/heart m^6^A-QTL for heat shock protein *HSPA4* involved in cardiac hypertrophy and fibrosis. Muscle/heart m^6^A-QTLs were enriched for 8 traits related to CVDs, reinforcing the roles of m^6^A regulators (*FTO*, *METTL3*, and *ALKBH5*) in CVDs. Li et al. utilized the quantitative trait loci of m6A, DNA methylation, and H3K27ac as genetic instruments to delineate locus-specific, directional maps of the crosstalk between m6A and the two fundamental epigenomic traits [104]. In muscle tissue, Li et al. uncovered 58 genetic loci that show strong multi-colocalization effects with the m^6^A-to-DNAme regulatory pairs, explaining GWAS traits such as high blood pressure and pulse rate. They also noticed that the variant rs12866090, associated with atrial fibrillation, likely mediates the crosstalk between m^6^A and DNA methylation in *CUL4A*.

Current evidence-based research unequivocally demonstrates that the integration of m6A with artificial intelligence and machine learning is highly significant for the prediction and diagnosis of CVDs. We anticipate that soon there will be more advanced methods with reduced error and false-positive rates, which will enhance our comprehension of the regulatory function of m6A in CVDs and offer new targets for forthcoming clinical interventions.

## 4. Discussion

This review provides a comprehensive summary of the pathophysiological effects of m6A methylation in cardiovascular risk factors, ischemia–hypoxia injury, atherosclerosis, myocardial infarction (MI), heart failure, and other cardiovascular diseases (Table 2 and Table 3). We also explored potential mechanisms of m6A methylation, including glucose metabolism, adipogenesis, VSMC differentiation and angiogenesis, autophagy, macrophage response, and inflammation, in the regulation of cardiovascular disease development.

However, m6A research is still in its early stages, and many unanswered questions remain. Currently, research on m6A methylation detection in cardiovascular diseases primarily focuses on the expression of *METTL3* and *FTO*. Limited attention has been given to the effects of m6A on the cardiovascular system through LncRNA, circRNA, and miRNA. This could be attributed to the relatively lower expression of methylated regulatory enzymes in the cardiovascular system compared to other systems (Figure 3). Future research should aim to investigate how other methylases and demethylases regulate the expression of downstream proteins under various pathological factors. Additionally, the role and mechanism of m6A-binding proteins in mediating the activity of m6A methylation in cardiovascular diseases should be explored.

Furthermore, CVDs are complex and multifactorial, involving multiple mechanisms. It is important to explore the synergistic effects of multiple methylated mRNA pathways on pathological progression. Dorn and Kmietczyk’s conflicting conclusions regarding the inhibitory effect of *METTL3* on cardiomyocyte hypertrophy suggest that changes in methylation patterns at different stages of disease occurrence and development may lead to distinct pathological processes. This highlights the need for further research on m6A-mediated methods for disease control, prevention, and treatment. Additionally, the effective clinical translation of m6A methylation findings is a topic that warrants comprehensive investigation [92,93].

## 5. Conclusions

M6A methylation participates extensively in diverse cardiovascular diseases. During the occurrence and progression of these diseases, proteins related to m6A methylation are also implicated in heart failure, myocardial hypertrophy, atherosclerosis, ischemic cardiomyopathy, and other conditions. It exerts a crucial regulatory function in the pathogenesis of these diseases. Moreover, the specificity of m6A varies. M6A methyltransferase can be applied in the early diagnosis of cardiovascular diseases. The dynamic changes in the expression and activity levels of *METTL3* and *METTL14*, as well as the m6A demethylase *FTO*, can serve as potential biomarkers for cardiovascular diseases.

## Figures and Tables

**Figure 1 medicina-61-00222-f001:**
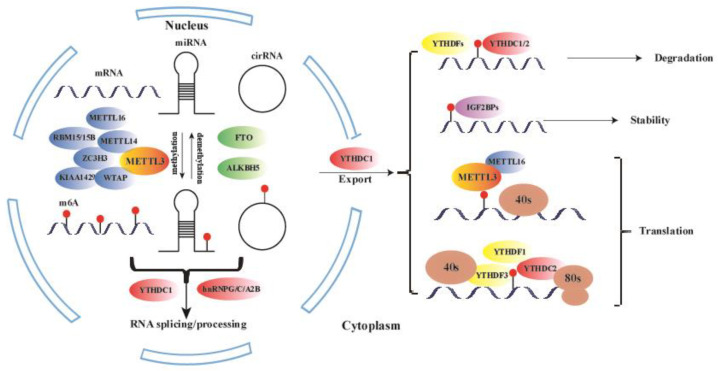
The regulation of m6A modification. m6A modification is established by m6A methyltransferases (“writers”) and removed by m6A demethylases (“erasers”). m6A-binding proteins (“readers”) recognize and bind to m6A-modified RNA, playing essential roles in RNA metabolism. *METTL3*, in collaboration with *METTL14/16*, *KIAA1429*, *WTAP*, *RBM15/15B*, and *ZC3H13*, forms the core methylation complex. This modification is reversible, with demethylases such as *ALKBH5* and *FTO* serving as m6A erasers. The modified transcripts are recognized by readers, including *YTHDF1/2/3*, *YTHDC1/2*, *IGF2BP1/2/3*, and *hnRNPG/C/A2B*, which subsequently influence various aspects of RNA function, such as translation promotion, stability, localization, splicing, and nuclear export.

**Figure 2 medicina-61-00222-f002:**
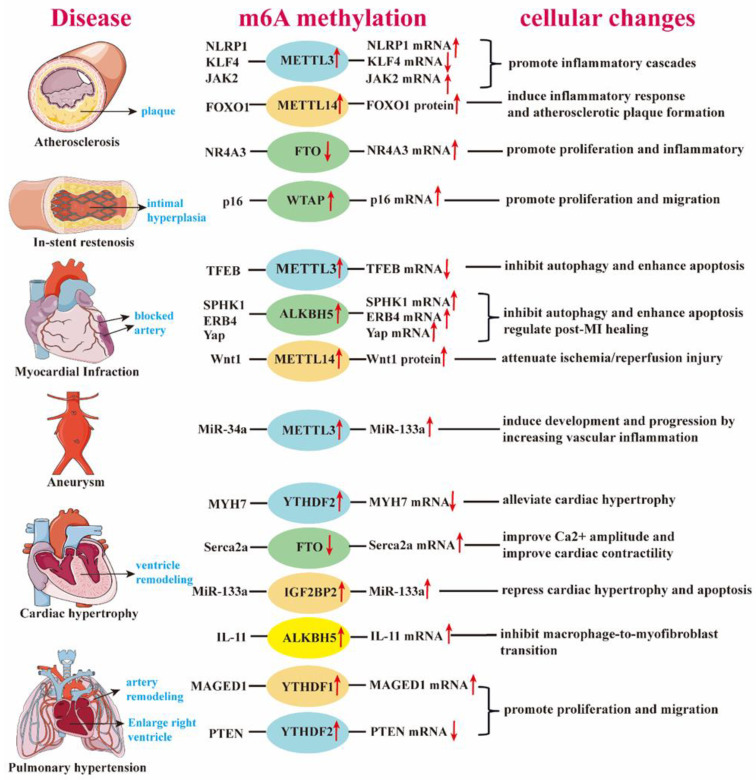
The role of m6A modulators in cardiovascular diseases and biological processes. *METTL3*-mediated m6A modification contributes to atherosclerosis by affecting inflammatory pathways, including *NLRP1*, *KLF4*, and *JAK2*. *METTL14* up-regulation induces inflammation and plaque formation by enhancing *FOXO1* translation. Reduced *FTO* expression promotes smooth muscle cell proliferation and inflammation by stabilizing *NR4A3* mRNA, accelerating atherosclerosis. In stent restenosis, *WTAP* promotes smooth muscle cell proliferation and migration by increasing *P16* mRNA via m6A modification. In myocardial ischemic disease, *METTL3* up-regulation decreases *TFEB* expression, impairing autophagic flux and enhancing cell apoptosis. During myocardial infarction repair, *ALKBH5* up-regulation inhibits autophagy by destabilizing mRNAs like *SPHK1*, *ERB4*, and *YAP*, promoting infarct repair. *METTL3* also mediates miR-34a maturation, which down-regulates *SIRT1* and promotes inflammatory infiltration in abdominal aortic aneurysm. Decreased *FTO* expression is linked to reduced *Serca2a* levels, leading to impaired cardiac contractility and heart failure. In heart failure, m6A-modified *MYH7* mRNA and miR-133a play protective roles in ventricular remodeling. Additionally, *ALKBH5* increases *IL-11* mRNA, inhibiting macrophage-to-myofibroblast transition. In hypoxic pulmonary hypertension, up-regulated *MAGED1* and down-regulated *PTEN*, both m6A-modified, contribute to smooth muscle cell proliferation, inflammation, and pulmonary vascular remodeling.

**Figure 3 medicina-61-00222-f003:**
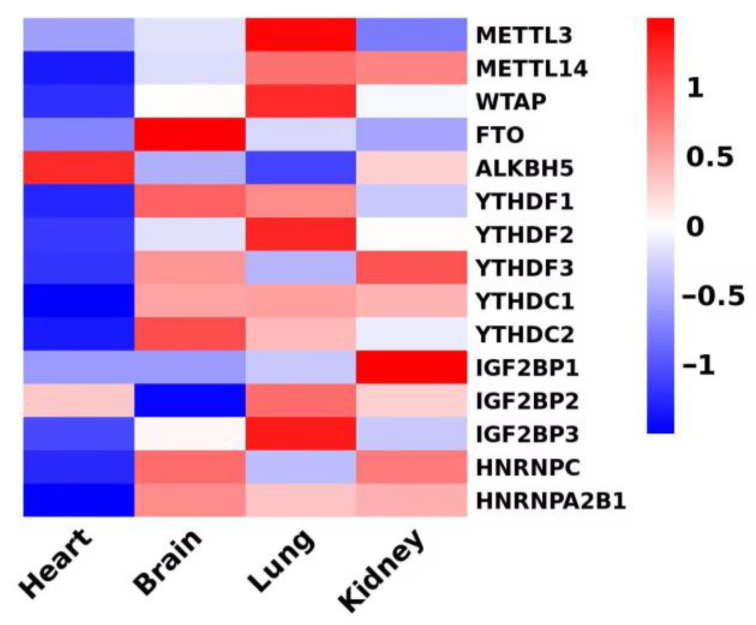
Relative amounts of methylated regulatory enzymes in the cardiovascular system and other systems. Heatmap shows differentially expressed methylated regulatory enzymes in different tissues. Data derived from PRJEB4337 of HPA RNA-seq normal tissues, in which RNA-seq was performed on tissue samples from 95 human individuals representing 27 different tissues in order to determine the tissue specificity of all protein-coding genes.

**Table 1 medicina-61-00222-t001:** Writers, erasers, and readers in m6A.

Type	Regulator	Function of RNA Modification	Reference
writers	*METTL3*	main catalytic subunit of m6A	[16]
	*METTL14/16*	activate *METTL3* through allosteric and RNA substrate recognition	[16]
	*WTAP*	the third subunit of the *METTL3-METTL14* complex	[17,18]
	*ZC3H3*	assist the localization of the methyltransferase complex in nuclear speckles and U-rich regions adjacent to the m6A sites in mRNAs	[19]
	*RBM15/15B*		[20]
	*KIAA1429*		[21]
erasers	*FTO*	demethylation of m6a	[2,22,23]
	*ALKBH5*		
readers	*YTHDF1*	promote mRNA translation	[24,25]
	*YTHDF2*	accelerate the decay of m6A-modified transcripts	
	*YTHDF3*	promote mRNA translation or enhance RNA decay	
	*YTHDC1*	promote mRNA translation and splicing and nuclear export	[26,27]
	*YTHDC2*	enhance translation	[28]
	*IGF2BP1/2/3*	regulate RNA localization, translation, and stability	[29]
	*hnRNPG/C/A2B*	promote RNA stability and mediate RNA splicing and microRNA process	[30]

Abbreviations: *ALKBH5*: ALKB homolog 5, RNA demethylase; *FTO*: FTO alpha-ketoglutarate dependent dioxygenase; *KIAA1429*: vir like m6A methyltransferase associated; *METTL3*: methyltransferase3; *METTL14*: methyltransferase14; *METTL16*: methyltransferase16; *RBM15/15B*: RNA binding motif protein 15B; *WTAP*: WT1associated protein; *YTHDF1*: YTH N6-methyladenosine RNA binding protein F1; *YTHDF2*: YTH N6-methyladenosine RNA binding protein F2; *YTHDF3*: YTH N6-methyladenosine RNA binding protein F3; *YTHDC1*: YTH N6-methyladenosine RNA binding protein C1; *YTHDC2*: YTH N6-methyladenosine RNA binding protein C2; *IGF2BP1/2/3*: insulin like growth factor 2 mRNA binding protein 1/2/3; *hnRNPG/C/A2B*: RNA binding motif protein X-linked G/C/A2B; *ZC3H3*: zinc finger CCCH-type containing3.

**Table 2 medicina-61-00222-t002:** Molecular mechanisms of m6a in risk factors in CVDs.

Risk Factors	Regulators	Cell	Regulation	Signaling	Function	Reference
glucose metabolism	*FTO*↑	hepatocellular cell	up-regulate mRNA	*FOXO1/FASN/* *G6PC/DGAT2*	improve the production of serum glucose and lipids	[35]
diabetes	*METTL14*↓	pancreatic β-cell	promote mRNA translation	*AKT/PDX1*	induce cell-cycle arrest and impair insulin secretion	[36]
obesity	*FTO*↑	endothelial cell	down-regulate mRNA	*AKT/* *prostaglandinD2*	aggravate vascular dysfunction	[37]
	*FTO*↑	preadipocyte	up-regulate mRNA	*JAK2* *-STAT3-C/EBPβ*	promote adipogenesis	[38]
	*FTO*↑	preadipocyte	control exonic splicing	*RUNX1T1*	modulate differentiation to promote adipogenesis	[39]
	*FTO*↑	preadipocytes	improve mRNA stability	*Atg5/Atg7*	promote autophagy and adipogenesis	[40]

Abbreviations: *FTO*: FTO alpha-ketoglutarate dependent dioxygenase; *METTL14*:methyltransferase14; *FOXO1*: forkhead box O1; *FASN*: fatty acid synthase; *G6PC*: glucose-6-phosphatase catalytic subunit; *DGAT2*: diacylglycerol O-acyltransferase 2; *AKT*: Akt kinase; *PDX1*: pancreatic and duodenal homeobox 1; *JAK2*: Janus kinase 2; *STAT3*: signal transducer and activator of transcription 3; *C/EBP*: CCAAT/enhancer binding protein; *RUNX1T1*: RUNX1 partner transcriptional co-repressor 1; *ATG5*: autophagy related 5; *ATG7*: autophagy related 7.

**Table 3 medicina-61-00222-t003:** The roles of m6A methylation in CVDs.

Diseases	Regulators	Cell	Regulation	Signaling	Function	Reference
calcification	*METTL3*↑	valve interstitial cell	up-regulate mRNA	*TWIST1*	promote osteogenic differentiation process	[44]
calcification	*METTL14*↑	smooth muscle cell	down-regulate mRNA	*Klotho*	promote calcification	[45]
Hypoxia-reoxygenation	*METTL3*↑	cardiomyocyte	down-regulate mRNA	*TFEB*	inhibit autophagy and enhance apoptosis	[46]
ischemic injury	*ALKBH5*↑	endothelial cell	up-regulate mRNA	*SPHK1/eNOS-AKT*	maintain angiogenesis	[47]
heart regeneration	*ALKBH5*↑	cardiomyocyte	improve mRNA stability	*YAP*	promote proliferation	[48]
post-ischemic	*ALKBH5*↑	endothelial cell	decrease mRNA stability	*WNT5A*	exacerbate dysfunction of CMECs	[49]
Hypoxia-reoxygenation	*FTO*↑	cardiomyocyte	up-regulate mRNA	*Mhrt*	inhibit apoptosis	[50]
Hypoxia-reoxygenation	*WTAP*↑	cardiomyocyte	up-regulate mRNA	*TXNIP*	enhance apoptosis	[51]
atherosclerosis	*METTL3*↑	endothelial cell	up/down-regulate mRNA	*NLRP1/KLF4*	promote inflammatory cascades	[52]
atherosclerosis	*METTL14*↑	endothelial cell	promote translation	*FOXO1/VCAM-1/ICAM-1*	induce inflammatory response and promote atherosclerotic plaque formation	[53]
atherosclerosis	*METTL3*↑	endothelial cell	up-regulate mRNA	*JAK2/STAT3*	promote atherosclerosis progression	[54]
atherosclerosis	*METTL14*↑	endothelial cell	up-regulate miRNA	pri-miR-19a /*DGCR8*	promote proliferation and invasion of ASVEC	[55]
atherosclerosis	*FTO*↓	smooth muscle cell	up-regulate mRNA	NR4A3	promote proliferation and inflammation	[56]
intimal hyperplasia	*WTAP*↓	smooth muscle cell	up-regulate mRNA	*p16*	promote proliferation and migration of VSMC	[57]
heart regeneration	*METTL3*↓	cardiomyocyte	up-regulate miRNA	miR-143 /*Yap*/*Ctnnd1*	inhibit heart regeneration	[58]
AML	*ALKBH5*↑	cardiomyocyte	not mention	TCA cycle	affect cell metabolism and survival	[59]
AML	*ALKBH5*↑	fibroblast	improve mRNA stability	*ErbB4*	regulate post-MI healing	[60]
Ischemia-reperfusion injury	*METTL14*↑	cardiomyocyte	promote translation efficiency	*Wnt1*	attenuate ischemia–reperfusion Injury	[61]
heart failure	*FTO*↓	cardiomyocyte	improve mRNA stability	*Serca2a*	improve Ca2+ amplitude	[62]
heart failure	*YTHDF2*↑	cardiomyocyte	promote mRNA degradation	*Myh7*	alleviate cardiac hypertrophy	[63]
heart failure	*IGF2BP2*↑	cardiomyocyte	promote miRNA accumulation on its target site	miR-133a	repress cardiac hypertrophy and apoptosis	[64]
heart failure	*FTO*↓	cardiomyocytes	improve mRNA stability	*Pgam2*	regulate glucose uptake	[41]
heart failure	*ALKBH5*↑	macrophage	improve mRNA stability	*IL-11*	inhibit macrophage-to-myofibroblast transition	[65]
pulmonary hypertension	*YTHDF2*↑	smooth muscle cell	promote mRNA degradation	*PTEN/PI3K/Akt*	enhance proliferation	[66]
pulmonary hypertension	*YTHDF1*↑	smooth muscle cell	promote translation efficiency	*MAGED1*	promote proliferation	[67]
aneurysm	*METTL3*↑	smooth muscle cell	up-regulate miRNA	*DGCR8*/ miR-34a	induce development and progression	[68]

Abbreviations: *ALKBH5*: ALKB homolog 5, RNA demethylase; *FTO*: FTO alpha-ketoglutarate dependent dioxygenase; *FOXO1*: forkhead box O1; *ICAM-1*: intercellular adhesion molecule 1; *JAK2*: Janus kinase 2; *METTL3*: methyltransferase3; *METTL14*:methyltransferase14; *Myh7*: myosin heavy chain 7; *MAGED1*: MAGE family member D1; *NLRP1*: NLR family pyrin domain containing 1; *KLF4*: KLF transcription factor 4; *NR4A3*: nuclear receptor subfamily 4 group A member 3; *SPHK1*: sphingosine kinase 1; *STAT3*: signal transducer and activator of transcription 3; *Serca2a*: Sarco/endoplasmic reticulum Ca(2+)-ATPase; *TFEB*: transcription factor; *WTAP*: WT1 associated protein; *Wnt1*: Wnt family member 1; *YAP*: Yes associated protein; *YTHDF1/2*: YTH N6-methyladenosine RNA binding protein F1/2.

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
