# Peer review of "Novel Insight of N6-Methyladenosine in Cardiovascular System"

_medicina, 2025, doi:10.3390/medicina61020222_

Round 1
Reviewer 1 Report
Comments and Suggestions for Authors
I have reviewed the manuscript entitled ‘Novel insight of N6-methyladenosine in cardiovascular system’.
The manuscript is well-presented and designed, it has significant contributions to the current literature for summarizing the current role of N6-methyladenosine.
The aim is well-shown and the target has been achieved.
I only recommend the presenting the potential role N6-methyladenosine in the era of artificial intelligence and ML. The potential role should be discussed citing ‘The Role of Artificial Intelligence in Coronary Artery Disease and Atrial Fibrillation’.
Author Response
Response to Reviewer 1 Comments
First, we want to express our gratitude to you! Big thanks for your kindly suggestions and sufficient patience! Our review is ” Novel insight of N6-methyladenosine in cardiovascular system”, ID: medicina-3369918. Thank you very much for your suggestions. They are very helpful to us.
Point1: I have reviewed the manuscript entitled ‘Novel insight of N6-methyladenosine in cardiovascular system. The manuscript is well-presented and designed, it has significant contributions to the current literature for summarizing the current role of N6-methyladenosine. The aim is well-shown and the target has been achieved.
Response1: Thank you!
Point2: I only recommend the presenting the potential role N6-methyladenosine in the era of artificial intelligence and ML. The potential role should be discussed citing ‘The Role of Artificial Intelligence in Coronary Artery Disease and Atrial Fibrillation.
Response2: We are very grateful to you for reviewing the paper so carefully. Thank you very much for your suggestion. We followed your suggestion and searched the relevant article about the potential role N6-methyladenosine in the era of artificial intelligence and machine learning. And we add relevant new results and developments to our manuscripts in title” 3.2.6. m6A in CVDs in the era of artificial intelligence and machine learning”. In this section, we summarize the application and latest scientific research results of m6A combined artificial intelligence in CVDs, including coronary artery disease, cardiac hypertrophy, high blood pressure and atrial fibrillation.
Reviewer 2 Report
Comments and Suggestions for Authors
Hello, thank you for the opportunity to evaluate this article.
I believe that the article is original, and the information is presented in a clear manner, respecting scientific rigor.
I would like to make a few suggestions to the authors:
1. Add a space before each bibliographic reference.
2. Remove the period before Introduction and Discussion.
3. Table 1 and Table 2: The first word in the title must be capitalized.
4. For Table 2, please add a legend explaining the abbreviations used in the table.
5. For 3.2, write m6A instead of m6a.
6. 3.2.2.1 METLL3 instead of Mettl3.
7. Please add a chapter related to Conclusions (they are missing in the article).
8. Genes must be written in italics. Please check and correct the entire text.
Author Response
Response to Reviewer 2 Comments
First, we want to express our gratitude to you! Big thanks for your kindly suggestions and sufficient patience! Our review is ” Novel insight of N6-methyladenosine in cardiovascular system”, ID: medicina-3369918. Thank you very much for your suggestions. They are very helpful to us.
Point1: Hello, thank you for the opportunity to evaluate this article. I believe that the article is original, and the information is presented in a clear manner, respecting scientific rigor.
Response1: Thank you!
Point2: Add a space before each bibliographic reference.
Response2: Thank you very much for your suggestion. We have added a space before each bibliographic reference.
Point3: Remove the period before Introduction and Discussion.
Response3: Thank you very much for your suggestion. We have removed two inappropriate periods before Introduction and Discussion.
Point4: Table 1 and Table 2: The first word in the title must be capitalized.
Response4: Thank you very much for your suggestion. We have capitalized the first word in the title in Table1 and Table2.
Point5: For Table 2, please add a legend explaining the abbreviations used in the table.
Response5: Thank you very much for your suggestion. We have added a legend explaining the abbreviations used in the table for Table 2.
Point6: For 3.2, write m6A instead of m6a.
Response6: Thank you very much for your suggestion. We have instead m6A of m6a in two places.
Point7: 3.2.2.1 METLL3 instead of Mettl3.
Response7: Thank you very much for your suggestion. We have instead METTL3 of Mettl3 in two places.
Point8: Please add a chapter related to Conclusions (they are missing in the article).
Response8: Thank you very much for your suggestion. We have added a chapter related to Conclusions, in which we summarized the role of m6A methylation in diverse cardiovascular diseases.
Point9: Genes must be written in italics. Please check and correct the entire text.
Response9: Thank you very much for your suggestion. We have corrected the wrong formatting and made sure that genes were written in italics.
Reviewer 3 Report
Comments and Suggestions for Authors
It is the reviewer's duty to point out errors:
Table 2 needs to be corrected:
Hepatocellular cell (not hepatoellular)
Pancreatic β-cell (instead of β-cell)
Please check the punctuation marks in the manuscript. Remove full stops in paragraph titles. Please note the correct positioning of text on each page.
This manuscript should be accepted for publication after minor revision.
Author Response
Response to Reviewer 3 Comments
First, we want to express our gratitude to you! Big thanks for your kindly suggestions and sufficient patience! Our review is ” Novel insight of N6-methyladenosine in cardiovascular system”, ID: medicina-3369918. Thank you very much for your suggestions. They are very helpful to us.
Point1: It is the reviewer's duty to point out errors
Response1: We are very grateful to you for reviewing the paper so carefully!
Point2: Table 2 needs to be corrected:Hepatocellular cell (not hepatoellular),Pancreatic β-cell (instead of β-cell).
Response2: Thank you very much for your suggestion. We have corrected misspellings in Table 2.
Point3: Please check the punctuation marks in the manuscript.
Response3: Thank you very much for your suggestion. We have checked the punctuation marks in the manuscript and corrected the wrong using of punctuation marks .
Point4: Remove full stops in paragraph titles.
Response4: Thank you very much for your suggestion. We have removed full stops in paragraph titles.
Point5: Please note the correct positioning of text on each page.
Response5: Thank you very much for your suggestion. We have noted the correct positioning of text on each page.
Point6: This manuscript should be accepted for publication after minor revision.
Response6: Thank you!